

**1** **Global aerosol typing classification using a new hybrid algorithm**

**2** **utilizing Aerosol Robotic Network data**

Xiaoli Wei[1,2], Qian Cui[4,5], Leiming Ma[1], Feng Zhang [2,3], Wenwen Li[2,3], Peng Liu[4]
[1] *Shanghai Meteorological Service 200030, China;*
[2] *Shanghai Qi Zhi Institute, Shanghai, 200232, China;*
[3] *Department of Atmospheric and Oceanic Sciences & Institute of Atmospheric Sciences, Fudan*
*University, Shanghai, 200438, China;*
[4] *School of Atmospheric Science, Nanjing University of Information Science and Technology, Nanjing*
*210044, China;*
[5] *Wuhan Meteorological Service 430000, China*
*Correspondence to: Feng Zhang (fengzhang@fudan.edu.cn)*

**12** **Abstract**

Aerosols have great uncertainty owing to the complex changes in their composition in
different regions. The radiation properties of different aerosol types differ
considerably and are vital in studying aerosol regional and/or global climate effects.
Traditional aerosol-type identification algorithms, generally based on cluster or
empirical analysis methods, are often inaccurate and time-consuming. Hence, we
aimed to develop a new aerosol-type classification model using an innovative hybrid
algorithm to improve the precision and efficiency of aerosol-type identification. An
optical database was built using Mie scattering and a complex refractive index was
used as a baseline to identify different aerosol types by applying a random forest
algorithm to train the aerosol optical parameters obtained from the Aerosol Robotic
Network sites. The consistency rates of the new model with the traditional Gaussian
density cluster method were 90%, 85%, 84%, 84%, and 100% for dust, mixed-coarse,
mixed-fine, urban/industrial, and biomass burning aerosols, respectively. The
corresponding precision of the new hybrid algorithm (F-score and accuracy scores)
was 95%, 89%, 91%, and 89%. Lastly, a global map of aerosol types was generated
using the new model to characterize aerosol types across the five continents. This
study utilizing a novel approach for the classification of aerosol will help improve the
accuracy of aerosol inversion and determine the sources of aerosol pollution.



**Keywords:** Aerosol typing classification, Hybrid algorithm, Complex refractive
index, AERONET

## 1. Introduction

Atmospheric aerosols are tiny solid or liquid particles suspended in the
atmosphere. Aerosols indirectly affect the energy budget and water cycle of the earth's
gas system by absorbing and scattering solar radiation or by changing the optical
properties and life cycle of the cloud as condensation nuclei of cloud droplets
(Redemann et al. 2000; Ramanathan et al. 2001). Additionally, desert dust, biomass
smog, and anthropogenic emissions of air pollutants can affect visibility, air quality,
and human health (Hess et al., 1998;Siomos et al., 2020;Tong et al., 2017) Evaluating
the impact of aerosols on radiative transfer is complex, primarily because of the
uncertainty of radiative forcing caused by the high spatiotemporal dynamic variation
of aerosol optical and physical characteristics in different regions (Kaskaoutis et al.,
2011;Che et al., 2018).The aerosol type embodies the long-term average
physicochemical properties of aerosols in a certain area (Kiehl & Briegleb, 1993;Lu et
al., 2023). Therefore, accurate identification of aerosol types can drive the study of the
climatic effects of aerosols, tracking and control of environmental pollution sources,
and precision of radiation transmission models.
Aerosol types are defined based on the radiation properties of different types of
aerosol particles owing to the large variation in their optical, physical, and chemical
properties. Currently, aerosol types are classified by two ways using two different
clustering techniques (Kumar et al., 2018). First, based on different sources and
properties at different observation points worldwide, aerosols are classified as
follows: dust aerosols from deserts, biomass combustion aerosols from forests or
grasslands, and urban/industrial (U/I) aerosols from fuel combustion in densely
populated urban areas (Dubovik et al., 2002;Pawar et al., 2015;Yousefi et al., 2020).
Second, based on the size of the radiation absorption rate, aerosols into four
categories: carbonaceous (fine-absorbing mode), soil dust (coarse absorption mode),



sulfates (nonabsorbing fine-grained mode), and sea salt aerosols (nonabsorbing
coarse-grained mode) (Kim et al., 2007;Levy et al., 2007). The second one is a type of
subcategorize anthropogenic aerosol. The first one is commonly used for aerosol
retrieval. Therefore, the first aerosol type classification is more common in research.
The optical properties of aerosols observed at ground stations are commonly used to
construct a two-dimensional identification space to obtain the aerosol types by
clustering techniques. Many combinations of optical properties and parameters are
available; They include $EAE_{440-870nm}$ (extinction angstrom exponent) vs. $SSA_{440nm}$
(single-scattering albedo), $AAE_{440-870nm}$ (absorption angstrom exponent) vs. $EAE_{440-870nm}$,
$AAE_{440-870nm}$ vs. $FMF_{550nm}$ (fine mode fraction), and $SSA_{440nm}$ vs. $EAE_{440-870nm}$
(Lee et al., 2010;Shin et al., 2019;Choi, et al., 2021). Studies have highlighted the
importance of selecting appropriate aerosol properties for accurate aerosol type
identification (Giles et al., 2012; Che et al., 2018).
Among the aerosol-type classification methodologies developed, those using
threshold and empirical analyses have the greatest potential for large-area and fixed-
period applications (Eck et al., 1999; Omar et al., 2005; Yang et al., 2009).
Traditionally, the aerosol-type classification algorithm mainly distinguishes different
aerosol types based on their optical properties and determines the threshold of their
optical properties based on clustering. However, the composition of aerosols changes
rapidly with time and location, owing to the combined influence of natural conditions
and human activities (for example, tornadoes and various anthropogenic activities)
(Sheridan et al., 2001). Unfortunately, determining aerosol types accurately and
rapidly is a challenge when using traditional methods (Bahadur et al., 2012;Shin et al.,
2019;Lin et al., 2021). Nevertheless, with advancements in data science, artificial
intelligence techniques have aided the accurate and rapid recognition of different
aerosol types.
Artificial intelligence algorithms can receive multiple aerosol characteristic
parameters as input, thus preventing the sole reliance of aerosol classification on a
limited number of features (Li et al., 2022). For example, Boselli (2012) performed a
k-means clustering analysis of single scattering albedo (SSA), aerosol optical depth



(AOD), electrical asymmetry effect (EAE), and asymmetry parameter (g) datasets for
the central Mediterranean Sea for the classification of aerosol into four: dusty,
continental, oceanic, or mixed aerosols. Nicolae (2018) developed a neural network
algorithm to estimate the aerosol typing of Lidar data and Hamill (2016) introduced
the Mahalanobis Distance for aerosol classification to determine a specific aerosol
type for each reference cluster. Li (2022) generated spatial contiguous aerosol type
map in China with an empirical aerosol type retrieval algorithm.Overall, limited
information on the optical properties of aerosols can reasonably determine the type of
aerosol (Hamill et al., 2016). However, some challenges remain in identifying aerosol
types through machine learning. First, the amount of valid ground aerosol property
data that can be used for training is less due to cloud removal and quality control.
Second, the accuracy of machine learning depends on the labeled aerosol typing
dataset, and finding a suitable classification method to classify the dataset is
challenging. Third, evaluating the accuracy of the final trained model is also tedious
(Zhang & Li, 2019;Siomos et al., 2020; Choi, et al., 2021a,b)
The traditional aerosol type identification methods are easily limited by time and
space, and most of them only classify aerosol types using two optical property
parameters, limiting the complete characterization of aerosols. Considering these
limitations, we aimed to (1) develop a new algorithm that can accurately and quickly
identify aerosol types to overcome existing problems such as low accuracy,
insufficient data, and difficulty in setting labels; (2) investigate the characteristics of
the regional spatial distribution of global aerosol types obtained using the new
machine learning algorithms, considering the large regional differences in aerosol
types. To achieve this, we propose a new aerosol-type classification algorithm based
on a Gaussian cluster and random forest algorithm to generate an aerosol-typing map
over several representative regions of the world.

**2. Study area and data**

Figure 1 shows the study area and the Aerosol Robotic Network (AERONET) site
distribution, which covers major regions of the world, to ensure the generalizability of




the research algorithm. We used 47 aerosol sites as marked on the map that were
distributed over five continents to train and verify machine learning by literature
review. The 47 sites represent different aerosol-type properties of different aerosol
source regions, including dust, mixed (mixed coarse and mixed fine aerosols), U/I,
and biomass-burning (BB) aerosols (Table 1 and Figure 1). Marine aerosols were not
considered because their low optical thickness values (generally <0.4) can result in a
less valid data scale that would not meet the study requirements. Here, the aerosol
source region refers to the area affected by one dominant emission source, where the
aerosol types are fixed and not easily confused (Giles et al., 2012;Hamill et al., 2016).
Table 2 presents the optical properties and microphysical characteristic parameters of
aerosols at four bands of AERONET (440, 675, 870, and 1020 nm). These parameters
were used to construct a database of SSA, AOD, and asymmetry parameters. Further,
typical sites dominated by different aerosol types worldwide were selected for
compositional analysis using the new model. The selected sites are distributed across
different regions of the world and represent a specific aerosol-dominated type and
aerosol source region.

For dust aerosols, five AERONET sites, namely Banizoumbou, Cape Verde,
Dakar, and Ouagadougou in Africa and Solar Village in West Asia, influenced by the
Saharan Desert, were considered. The Dakar and Cape Verde sites are located at the
tip of the Cape Verde Peninsula—the westernmost part of Africa, bordering the
Atlantic Ocean. Although these two sites are located in the ocean, they are dominated
by dust aerosols influenced by aerosol plumes in the Saharan Desert. Moreover, the
Banizoumbou and Ouagadougou sites are in the middle of Africa. Here, the
northeasterly winds prevail in winter, and northwesterly winds prevail in summer,
which can bring dust aerosols from the Saharan Desert. For mixed aerosols, the
AERONET sites Ilorin, Kanpur, Sede Boker, and XiangHe were selected. For U/I
aerosols, the AERONET sites GSFC, Ispra, Mexico City, and Moldova were selected.
Four AERONET sites, namely, AltaFloresta, Abracos Hill, LakeArgyle, and Mongu,
were selected as BB aerosol-dominant sites





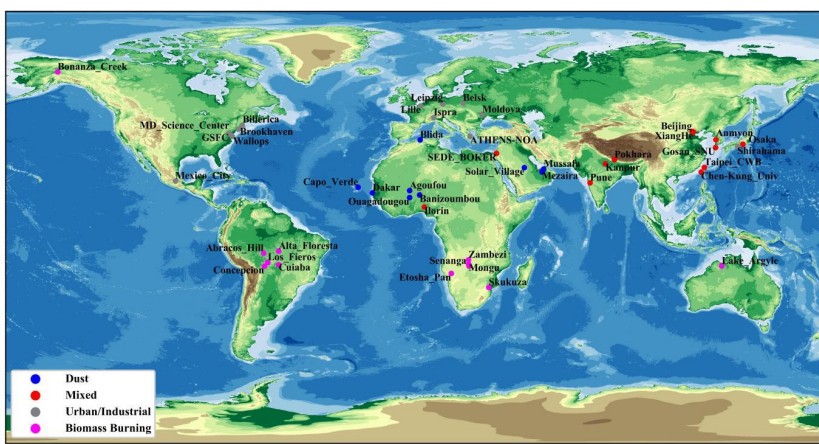

**Figure 1**. Study area and 47 AERONET sites selected by literature review.

**Table 1**. 47 AERONET sites selected by literature review.

| Aerosol Type | Sites for Training | Sites for Testing |
|---|---|---|
| Dust | Agoufou,Capo_Verde,Dakar, Mezaira, Mussafa, Ouagadougou, | Banizoumbou,Solar_Village, Blida |
| Mixed | Anmyon,Beijing, Chen-Kung_Univ,Ilorin,Kanpur, Sede_Boker, Gosan_SUN, Pune, Taipei_CWB | Osaka, XiangHe, Pokhara, |
| Urban/Industry | Brookhaven,Billerica,Belsk,GSFC,Ispra,UMBC,Lille, Mexcio_City,Moldova,MD_Science_Center,Wallops | Athens-Noa,Shirahama, Leipzig |
| Biomass Burning | Abracos_Hill,Alta_Floresta,Cuiaba,Concepcion Los_Fieros,Mongu,Senanga, Skukuza,Zambezi | Bonanza_Creak, Etosha_Pan, Lake_Argle |

**Table 2**. The optical and microphysical properties for aerosol type identification.

| | Parameters | Variables (band waves) |
|---|---|---|
| Optical Properties | Ångström Exponent (AE) | EAE (440-870)[1] |
| | Aerosol Optical Depth (AOD) | AOD (440,675,870,1020)[1] |
| | Single Scattering Albedo (SSA) | SSA (440,675,870,1020)[1] |
| | Asymmetry Parameter | g (440,675,870,1020)[1] |
| | Imaginary Part of the Complex Refractive Index | REFI (440,675,870,1020)[1] |
| | Real Part of the Complex Refractive Index | REFR(440,675,870,1020)[1] |
| Microphysical Properties | Effective Radius | EffRad-F[2], EffRad-C[2] |
| | Standard Deviation of Effective Radius | StaDev-F[2], StaDev-C[2] |
| | Size Distribution | Vol-Con (0.05-15μm) |

Note:[1] refers to wavelength in nm; [2] refers to different modes; EAE is Extinction Ångström Exponent; REFI is Imaginary Part of the

Complex Refractive Index; REFR is Real Part of the Complex Refractive Index; F refers to fine mode; C refers to coarse mode; EffRad is

Effective Radius; StaDev is standard deviation; Vol-Con is Volume concentration



## 3. Methods

A new aerosol classification typing hybrid approach that provides insight into spatiotemporal variations in aerosol pollution and climate impacts on a global scale is proposed in this study. In this approach, an aerosol optical properties database using the Mie scattering model was built for calculating rapidly unique aerosol-type features. Additionally, the approach introduced, for the first time, the median value of the complex refractive index (CRI) as the criterion for identifying the aerosol type. Further, we have selected the aerosol classification based on the source (as described in Section 1), according to the parameters applied in this study and the requirements for AOD retrieval. Figure 2 shows the working flowchart of the hybrid aerosol-type identification approach, including three stages: aerosol typing preliminary classification, aerosol optical database generation, and global aerosol typing identification and validation. The details of these three stages are as follows.

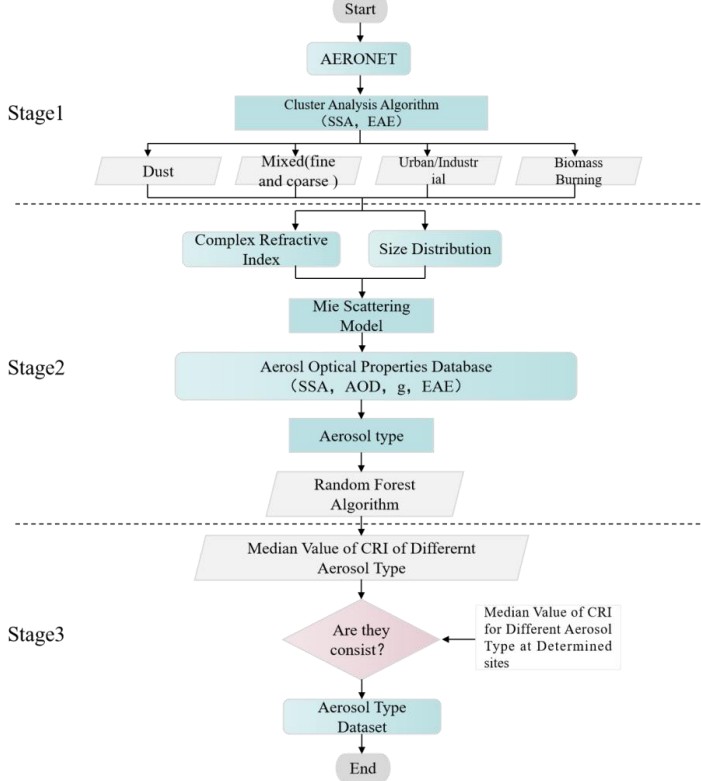

**Figure 2**. Flow chart of the new hybrid algorithm in aerosol type identification.





### 3.1 Aerosol typing preliminary classification (Stage 1)

Stage 1 aimed to solve the problem of obtaining a feature parameter dataset for the baseline aerosol type. In previous studies, the Gaussian kernel density clustering method showed great potential for distinguishing the optical properties of different aerosol types and determining their corresponding thresholds rapidly ( Kalapureddy et al. 2009;  Pathak et al. 2012). The high concentration value in each cluster generally represents the dominant pattern of a specific aerosol type, particularly the data within the window, taking the cluster centroid as the center and a specific distance as the radius. Preliminary aerosol-type datasets can be generated by digging deep into the distribution information of the effective radius, variance, and refractive index of the data within the window. The spectral absorbability and particle size of aerosols guide the identification of dust, carbonaceous, or hygroscopic aerosols; SSA indicates the absorption of aerosol particles; and EAE describes aerosol particle size (Giles et al., 2012). Consequently, in this study, $SSA_{440nm}$ and $EAE_{440-870nm}$ of 47 AERONET sites and the Gaussian kernel density method were used to estimate the relative densities and determine the primary patterns of the dominant aerosol types; here, the aerosol type was classified as a dust aerosol. Eqs. (1) and (2) represent the kernel density and Gaussian kernel density methods (Rosenblatt, 1956)

$$f_{X(v)} = \frac{1}{L}\sum_{i=1}^{L} k_\sigma (\frac{\vec{x}-\vec{x}_i}{\sigma}) \ , \qquad (1)$$

where $f_{X(v)}$ denotes the kernel density and $k_\sigma$ indicates the kernel function. $x_1$, $x_2$... $x_L$ are the sample points of independent identical distribution. Mathematically, kernel functions are symmetric, normalized, and sample-centric when used for density estimation; this is best described by the Gaussian kernel equation given by Eq. (2).

$$k_\sigma = \frac{1}{\sqrt{2\pi}\sigma} \exp(\frac{-|\vec{x}-\vec{x}_i|^2}{2\sigma^2}) , \qquad (2)$$

where σ is the kernel size used as a smoothing factor (Moraes et al., 2021).

The mixed aerosols comprised fine- and coarse-mode aerosols, indicated by EAE > 0.8 and EAE ≤ 0.8, respectively. Figure 3 shows the clustering distribution of



EAE and SSA using the Gaussian kernel density method for different aerosol types at
the 47 AERONET sites. For the dust aerosol cluster, the density core area EAE was
0.1–0.3, and SSA was 0.89–0.94, implying that it contained many coarse aerosol
particles with moderate absorptivity. Furthermore, the mixed aerosols had two distinct
centers: one for the coarse-mode aerosols with a median EAE value of 0.4, indicating
that the cluster contained massive high-absorption aerosols, and the other for fine-
mode aerosols with a median EAE value of 1.3. Low-absorption aerosols were
dominant in the cluster, similar to U/I aerosols. Additionally, the density core region
EAE of U/I aerosol was 1.5–1.8, and SSA was 0.94–0.97, implying the dominance of
fine and low-absorption aerosols. Conversely, BB aerosols had two indistinct centers.
This is because, during biomass combustion, gas and particulate matter emissions are
limited by the combustion conditions, divided into combustion and simmering.
Combustion produces black smoke, and simmering produces white smoke.
Combustion, such as burning flames (grass) with high black carbon content, has a
strong absorption capacity, resulting in a low SSA. Simmering, such as burning wood
(i.e., trees), tends to be smoldering, lasts longer, has a weaker absorption capacity, and
has a higher SSA value. Therefore, despite possessing different absorption
characteristics, BB aerosols are defined as one aerosol type with an unseparated
center of combustion and simmering.





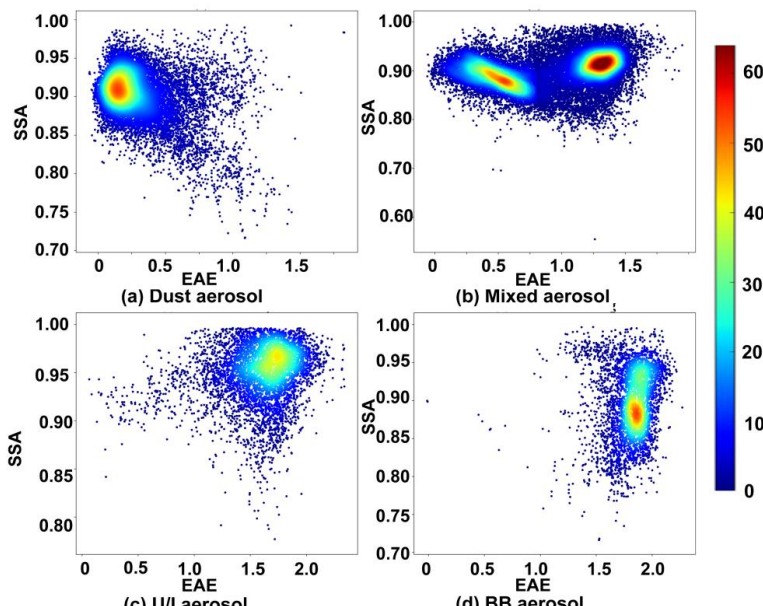

**Figure 3.** The clustering distribution of EAE and SSA using the Gaussian kernel density method for different aerosol types.

## 3.2 Aerosol optical database generation (Stage 2)

In stage 2, the aerosol optical parameter database was built using the aerosol size distribution parameters, CRI, and Mie scattering model. The main reasons for constructing an aerosol optical parameter database instead of using the AERONET data directly are as follows: 1) many data are missed in AERONET, particularly those for sites dominated by biomass combustion, which does not meet the requirements of machine learning methods or traditional aerosol type identification algorithms; 2) Calculating the optical properties of aerosols based on a fixed refractive index can accurately determine aerosol types. Therefore, once the aerosol spectral distribution parameters, such as effective radius, variance, and refractive index of the five aerosol types, are determined in stage 1, the aerosol optical parameter database can be constructed using the Mie scattering model in stage 2, assuming that aerosols are spherical particles. The Mie scattering model is a simple, practical, and ideal spherical particle model commonly used in radiation transport models (Michael et al., 1994). Figure 4 shows the details involved in the building aerosol optical database. The



aerosol optical database has four major parameters (AOD, EAE, SSA, and g) at four
wavelengths (440, 675, 870, and 1020 nm, respectively).




**Figure 4**. The diagram of building aerosol optical database.

As shown in Figure 4, size distribution is a major parameter in building aerosol

optical databases. Table 3 presents the aerosol size distribution parameters, including
the effective radius and standard deviation range for the five aerosol types in the
coarse and fine modes, which were calculated using the data in the window
determined by the Gaussian kernel density algorithm. These aerosol size distribution
parameters were used to build the aerosol optical database for the Mie scattering
model.
**Table 3**. Size distribution parameters of five aerosol types in coarse and fine mode (unit: μm)

| Aerosol type | REFF-fine | REFF-coarse | Std-fine | Std-coarse |
|---|---|---|---|---|
| Dust | 0.05-0.42 | 1.3-2.65 | 0.5-0.8 | 0.4-0.7 |
| Mixed-coarse | 0.05-0.25 | 1.25-3.5 | 0.4-0.8 | 0.4-0.7 |
| Mixed-fine | 0.05-0.27 | 1.2-4.5 | 0.3-0.6 | 0.5-0.8 |
| U/I | 0.05-0.26 | 1.45-3.5 | 0.3-0.6 | 0.5-0.8 |
| BB | 0.05-0.17 | 1.35-4.5 | 0.3-0.5 | 0.5-0.8 |

The Mie scattering model has various size distribution functions, including log-

normal, power-law, and bimodal log-normal distributions, which describe the aerosol
type. According to the particle radii provided by AERONET, the size distributions of





different aerosol types can be divided into coarse and fine modes. The bimodal log-
normal function [Eq. (3)] is reportedly the most suitable size distribution function for
modeling aerosol particle size distribution (Remer et al., 2009):
$$n(r) = cons\tan t \times r^{-4}\{\exp(-\frac{(\ln r - \ln r_{g1})^2}{2\ln^2 \sigma_{g1}}) + \gamma \exp(-\frac{(\ln r - \ln r_{g2})^2}{2\ln^2 \sigma_{g2}})\} \ , \quad (3)$$
where n(r) is the number of particles at different radii; constant is obtained by fitting;
While $r_{g1}$ and $r_{g2}$ denote the radii, $\sigma_{g1}$ and $\sigma_{g2}$ denote the variances of the aerosol in the
coarse and fine modes, respectively; and $\gamma$ is determined by the volume distribution.
In the bimodal normal distribution model, $\gamma$ is the ratio of coarse to fine modes, which
can be fitted by the volume distribution from AERONET; notably, volume distribution
is the average of the standard aerosols obtained after clustering at the training sites.

Figure 5 shows the volume distributions of the different aerosol types. The

aerosol volume distribution of dust aerosol-dominant sites focuses on the large radius;
the peak value of $\gamma$ was 8.1, and the radius of dust aerosols was 1.5–2.0 μm.
Additionally, the mixed-coarse aerosol with the radius in the range of 0.04–0.2 μm
and 4.9 as the maximum value of $\gamma$. The mixed-fine aerosol had two obvious peaks:
one with a large radius, namely the coarse mode, with a radius of 2.2–3 μm and 2.1 as
the peak point of $\gamma$; a second with a small radius of 0.1–0.22 μm and 0.14 as the peak
point of $\gamma$. Moreover, the volume distributions of U/I and BB aerosols were similar.
Both had a relatively low range of $\gamma$ values at large radii and relatively high values at
small radii, with peak values of 0.81 and 0.7 for U/I and BB aerosols, respectively.



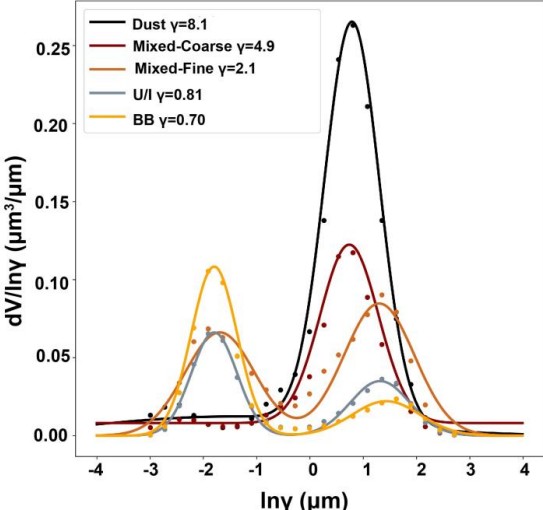

**Figure 5.** Volume distribution of the five aerosol types.

The CRI of aerosols is another key parameter among aerosol optical properties; it determines inherent optical properties of aerosols, such as scattering and absorption (Raut and Chazette, 2008). The CRI is vital for determining aerosols' chemical and physical compositions (Dubovik and King, 2000). Aerosols in the real atmosphere are usually mixed with different types of particles, which a single refractive index cannot identify; however, the CRI represents the entire aerosol model in the atmosphere (Redemann et al., 2000). Ideally, the CRI and aerosol components can be mutually determined (Wu et al., 2021). Table 4 depicts the CRI standard values for the five aerosol types obtained by calculating the median value of the CRI of the dominant aerosol type after Gaussian density clustering. These values were used as a baseline for identifying the aerosol types in subsequent studies. As presented in Table 4, the minimum imaginary index part is represented by the dust aerosol with CRI of 0.00374, 0.000847, 0.000847, and 0.000820 at 440, 675, 870, and 1020 nm, respectively, owing to the weakest absorption of dust aerosols. Moreover, the imaginary index part of the mixed-fine aerosols (0.01) was close to that of the U/I (0.07) and BB aerosols (0.02) because of their similar absorption properties.




**Table 4**. Real and imaginary index of CRI for the five aerosol types (Bands:440/675/870/1020
nm).

| Aerosol Type | Imaginary Index | Real Index |
|---|---|---|
| Dust | 0.00374/0.000847/0.000847/0.000820 | 1.4671/1.4778/1.4622/1.4504 |
| Mixed-coarse | 0.005349/0.002444/0.00204657/0.001845 | 1.4672/1.5088/1.5049/1.495 |
| Mixed-fine | 0.01449/0.01001/0.01009/0.009589 | 1.5075/1.5203/1.5243/1.516 |
| U/I | 0.007185/0.007166/0.007536/0.007552 | 1.4497/1.4397/1.4383/1.4346 |
| BB | 0.01961/0.01906/0.01903/0.01854 | 1.5133/1.5261/1.531/1.5282 |

Lastly, by fixing the CRI, changing the size distribution, and using the Mie
scattering model, we generated the aerosol optical property database for five aerosols,
including the data for AOD, EAE, SSA, and g. In the aerosol optical property
database, AOD is the value obtained after eliminating the influence of the aerosol
concentration. The AOD was obtained from the extinction cross section ($C_{ext}$)
calculated using the Mie scattering model in Eqs. (3) and (4), where $\beta_e$ is the
extinction coefficient, n(r) is the aerosol spectral distribution, and N(z) is the variation
of aerosol concentration with height. Notably, the effect of aerosol concentration
needs to be removed from the AOD when referring to aerosol optical properties. The
AOD was normalized by dividing the aerosol optical thickness at the four
wavelengths by the optical thickness at 440 nm. The other parameters (EAE, SSA,
and g) were calculated using Eqs. (6) – (8).
$$\beta_{e/s} = \int_{r\min}^{r\max} C_{ext/sca} n(r) dr \quad , \tag{4}$$

$$\tau_{e/s} = \int_0^{Z_{top}} \beta_{e/s} N(z) dz \quad , \tag{5}$$

$$EAE_{440-870nm} = -\frac{\ln(\tau_{440nm}) - \ln(\tau_{870nm})}{\ln(440) - \ln(870)} \quad , \tag{6}$$

$$SSA = \frac{\tau_s}{\tau_e} \quad , \tag{7}$$

and
$$g = <\cos\Theta> = \frac{1}{2} \int_{-1}^{1} p(\cos\Theta) \cos\Theta \, d\cos\Theta \quad , \tag{8}$$



where $\tau_{440}$ and $\tau_{870}$ are the extinction optical depths of the aerosol at 440 and 870 nm,
respectively, $EAE_{440\text{-}870}$ nm is the extinction Ångström index from the 440 to 870 nm
band, and Θ denotes the scattering angle.
The amount of data for the five aerosol types calculated using the Mie scattering
model is presented in Table 5. The least amount of data was observed for the mixed-
fine aerosol owing to its small distribution range of effective variance. The largest
data was observed for dust and mixed aerosols owing to their widely distributed
effective radii. A total of 326400 datasets were present in the aerosol optical database,
which meets the requirements for random forest algorithm.
**Table 5**. The data size of optical database simulated by Mie scattering model.

| Total | Dust | Mixed-coarse | Mixed-fine | U/I | BB |
|---|---|---|---|---|---|
| 326400 | 88200 | 96000 | 42000 | 51840 | 48360 |

**3.3 Global aerosol type identification and validation (Stage 3)**
In stage 3, the random forest model was introduced to the aerosol-type
identification algorithm. The random forest model is an integrated model based on
classification and regression trees, in which multiple trees are aggregated using
majority voting and averaging for classification and regression (Breiman, 2001). The
model has a high prediction accuracy, excellent tolerance for abnormal values and
noise, and a hard overfit. In a comparison by Fernandez (2014), the random forest
algorithm performed the best among 179 classification algorithms.
In this study, the input parameters for random forest model training, including
$SSA_{440nm}$, $SSA_{675nm}$, $SSA_{870nm}$, $SSA_{1020nm}$, $g_{440nm}$, $g_{675nm}$, $g_{870nm}$, $g_{1020nm}$, normalized
$AOD_{440nm}$, $AOD_{675nm}$, $AOD_{870nm}$, $AOD_{1020nm}$, and $EAE_{440\text{-}870nm}$, were selected from the
aerosol optical property database, and the expected output values were the specific
aerosol types. The random forest model was optimized and the parameters were
determined using the grid-searching method. The parameters, including n_estimators
(classifier), max_features (maximum feature value), and min_samples_leaf (minimum
number of samples for nodes), were set as 160, 10, 12, and 12, respectively. Then,
based on the trained and optimized model, aerosol typing of any AERONET site in



different regions of the world can be identified quickly. Generating the aerosol type
distribution map on a global scale is vital for regional and global climate studies and
ground remote sensing.
**4 Results**
**4.1 Algorithm comparison**
To demonstrate the effectiveness of the new hybrid algorithm, its performance
was compared with that of Gaussian density clustering algorithm. Figure 6 shows the
confusion matrix between the new hybrid and Gaussian density clustering algorithms
in identifying aerosol types. The results of the hybrid algorithm showed 90%
consistency with that from the Gaussian density clustering algorithm, in delineating
dusty aerosols, indicating that its efficiency in identifying dust. For mixed-coarse
aerosols, the consistency reached 85%, with 14% identified as mixed-fine aerosols,
1% as dust by the hybrid algorithm, and 15% as mixed-coarse aerosols by the
Gaussian density clustering algorithm. Similarly, for mixed-fine aerosols, both
algorithms showed 84% consistency, with 14% identified as a mixed-coarse aerosol
by the hybrid algorithm and as a mixed-fine aerosol by the Gaussian density cluster
algorithm. Furthermore, both algorithms identified 84% of U/I aerosols correctly, with
the remaining 16% identified as mixed aerosols (fine and coarse). Lastly, the
classification of BB aerosols using these two methods was the same. Overall, the
Gaussian density clustering and hybrid algorithms were consistent in dust, mixed-
coarse, U/I, and BB aerosol identification.



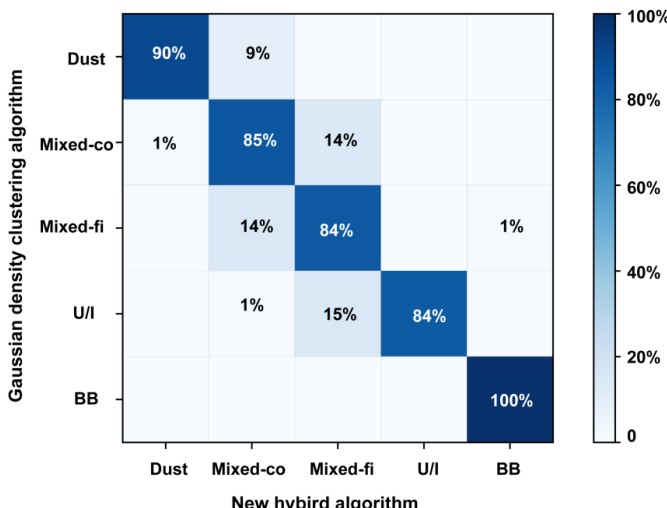


**Figure 6**. The confusion matrix between Gaussian density clustering and hybrid algorithm.
As described in the Methods section, a specific aerosol type theoretically has a
fixed CRI owing to its constant composition. The CRI characterizes the mixture
composition of aerosol particles and is a key parameter controlling the inherent
scattering and absorption characteristics of aerosol particles. To further analyze the
accuracy of the new algorithm, the aerosol CRI was applied as a key criterion for
aerosol identification. The CRI has two parts: imaginary and real. The imaginary part
indicates radiation absorption by aerosols, with a small value signifying a small
absorption. Because the radiation of aerosols is more dependent on the imaginary than
the real part, the imaginary part is essential for inferring the optical properties and
aerosol types. Hence, we compared the real and imaginary parts of the CRI calculated
using the hybrid and Gaussian density clustering algorithms.
Figure 7 shows box plots of the aerosol CRI for dust, mixed-coarse, mixed-fine,
U/I, and BB aerosols using the hybrid classification and Gaussian density clustering
algorithms. Based on the principle that the CRI of aerosols is fixed under ideal
conditions, the closer the median value of the CRI of the identified aerosol type is to
the median value of the benchmark CRI, the more accurate is the identification
method.



As shown in Figures 7 (a) and (f), the median values of the CRI real part for dust
aerosol are in the range 1.45–1.53 at four bands, and those of the imaginary part are
0.003–0.004 at 440 nm; further, the values in other bands decrease rapidly as
wavelength increases. The imaginary part of CRI represents the absorption of light by
the aerosol, with a small absorption indicating strong scattering. The results of the
imaginary part are consistent with the spectral dependence properties of dust-based
aerosols according to the wavelength. This is primarily because dust aerosols,
composed of clay, quartz, and hematite, exhibit strong absorption in the blue band
(440 nm) and low absorption in the visible and near-infrared bands. For the dust
aerosols, the CRI determined by the two methods did not differ much. However, the
median value of the CRI obtained using the hybrid algorithm was slightly closer to the
benchmark CRI than that obtained using the Gaussian density clustering algorithm for
dust aerosols. Therefore, the hybrid algorithm was concluded to be more accurate in
identifying dust aerosol.
Figures 7 (b) and (g) show the median values of the CRI real part for mixed-
coarse aerosol is 1.47–1.55 at four bands using the new hybrid algorithm, but the
imaginary part is 0.004–0.009 at 440 nm. However, the real part is 1.44-1.50 at four
bands determined by Gaussian density clustering algorithm, and the imaginary part is
0.006–0.009 at 440nm. The median value of the hybrid algorithm was closer to the
baseline median value than that of the Gaussian density clustering algorithm for both
the real and imaginary parts.
Figures 7 (c) and (h) show the median value of the CRI real part for mixed-fine
aerosols determined using the new hybrid and Gaussian density clustering algorithms,
which was 1.42–1.51 at four bands. This result is close to the range (1.44–1.52)
reported by Wu (2021) in Beijing using a random forest algorithm. The median CRI
of the real part at four bands and imaginary part at the (675-870-1020 nm) bands were
close to the baseline median value for the new algorithm. Additionally, the median
value of the imaginary part was lower than that of the new hybrid algorithm and
further from baseline data for the identifying aerosol type results mixed with 14%
coarse aerosols. Mixed coarse aerosols result in weaker absorption. Hence, the new





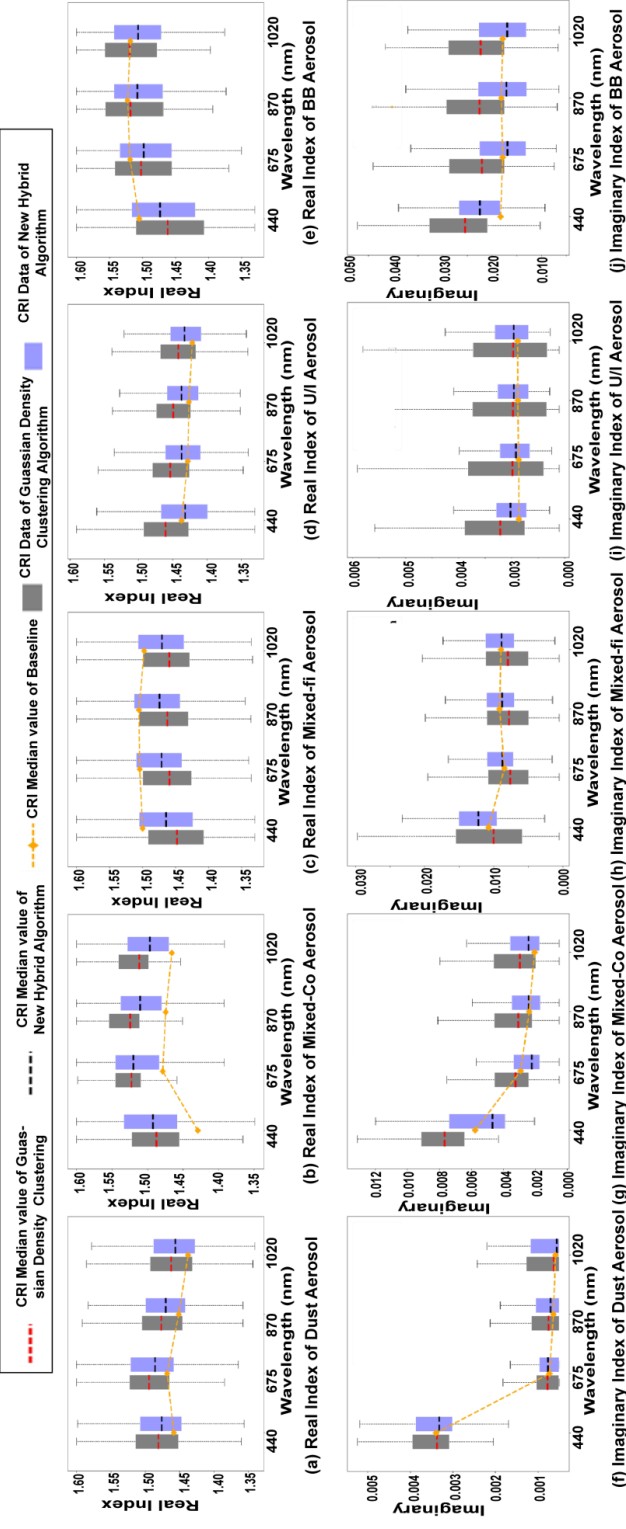

**Figure 7.** Box plots of the real index and the imaginary index of the CRI for (a) dust, (b)mixed-coarse, (c) mixed-fine aerosol, (d)U/I, and (e) BB aerosol identified by the Gaussian density clustering algorithm and new hybrid algorithm, respectively (the upper line is the real part, and the bottom line is the imaginary part).



hybrid algorithm performed better at identifying mixed-fine aerosols than the
Gaussian density clustering algorithm.

Similarly, as seen in Figures 7 (d) and (i), the median value of the CRI real part

for U/I aerosol identified using the new hybrid algorithm was 1.39–1.47. This median
value was lower than that of the mixed-fine aerosols. This is because the real part
indicates the absorption ability of aerosols, and the absorption ability of U/I aerosols
was less than that of mixed-fine aerosols. For the imaginary part also, the new hybrid
algorithm performed slightly better than the Gaussian density clustering algorithm at
the four bands.

For BB aerosols, the median value of the real part generated using the new hybrid

algorithm differed slightly from that generated by the Gaussian density clustering
algorithm. Additionally, the median obtained using the Gaussian density clustering
algorithm was closer to the baseline. Furthermore, when analyzing the imaginary part,
the new hybrid algorithm performed much better than the Gaussian density clustering
algorithm. Even with a 100% concordance rate between the new hybrid and Gaussian
density clustering algorithms in identifying BB aerosols, the refractive index still
differed. This result indicates that 1% of mixed-fine aerosols classified using the
Gaussian density clustering algorithm were correctly identified as BB aerosols by the
new algorithm. Overall, these results demonstrate that the new algorithm is reliable.

### 4.2 Aerosol type determination for typical sites

### 4.2.1 Dust aerosol

Figure 8 shows the aerosol types obtained using the new hybrid algorithm for the

five sites selected for dust aerosol identification. According to the prediction by the
new hybrid algorithm, the aerosols at these five sites mainly contained dust aerosols
along with a small amount of U/I, mixed-fine, and BB aerosols, and a large amount of
mixed coarse aerosols. This shows that other types of aerosols invaded these areas
besides dust aerosol. BB aerosols may have been transferred from the southern
African savannah. Additionally, U/I aerosols could be from industrial cities, such as
Dakar, Abidjan, and Lagos, which are dominated by anthropogenic aerosols and are





close to the AERONET sites.

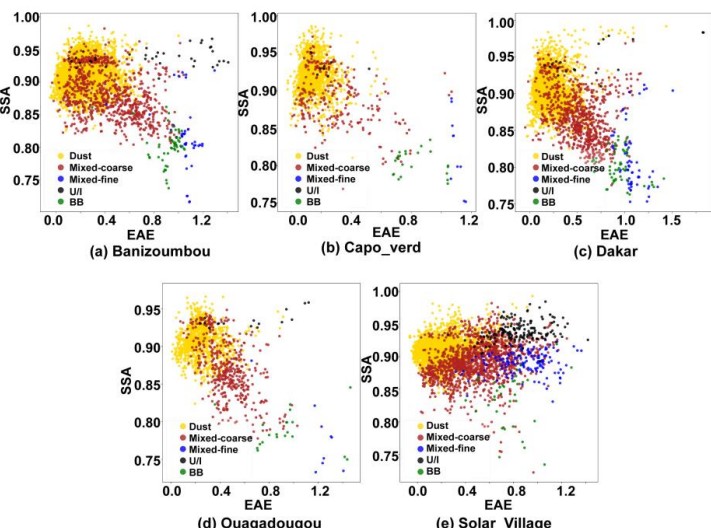


**Figure 8**. Identification of dust aerosol at dominant aerosol sites.
**4.2.2 Mixed aerosol**
Besides Ilorin in Africa, the mixed aerosol AERONET sites, including Kanpur,
Sede Boker, and XiangHe, are in Asia. The aerosol types at these four sites were
determined using the new hybrid algorithm (Figure 9). Mixed coarse aerosols
dominated the Kanpur, Ilorin, and Sede Boker sites, and mixed fine aerosols
dominated XiangHe. Part of the dust in Xianghe could be due to the Takla Desert in
spring and the westerly winds prevailing in western China, which transported dust
aerosols over long distances. Additionally, the U/I aerosol in Xianghe could be a result
of human activities, construction emissions, and fuel burning in winter. The BB
aerosol was traced to the burning of a small amount of biomass in Xianghe, located in
a suburban area.
Furthermore, excluding dust aerosols, we observed BB and U/I aerosols in the
Kanpur site in the Ganges Basin of India. A certain amount of U/I and dust aerosols
were also observed in Sede Boker, located in the industrial center of Israel, possibly
from the Arabian desert. Lastly, Ilorin had the most dust and least BB aerosols
because it is located in central Africa, often affected by the Saharan Desert and



African savannah.

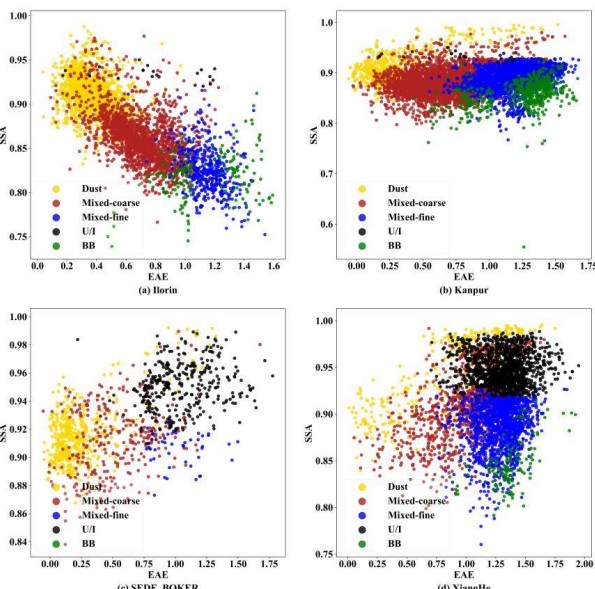


**Figure 9**. Same as Figure 8 but for Mixed aerosol.

### 4.2.3 Urban/industrial aerosol

All the selected AERONET sites for evaluating the performance of the new
hybrid algorithm in terms of U/I aerosol identification are in Europe or North America
(Figure 10). GSFC is located in the densely populated and industrially developed area
of Washington in the United States, explaining its complex aerosol type dominated by
the U/I aerosol followed by a few mixed and BB aerosols and a small amount of dust
aerosols.
Ispra is in Turin, one of Italy's largest industrial centers. However, dust-type
aerosols were identified, possibly transported from the Libyan desert when Italian
winters were controlled by southwesterly winds. Moreover, Mexico, where the
Mexico City site is located, is an industrialized country with modern industries and
agriculture, abundant oil production, and a dense population. Nevertheless, we
identified dust, mixed coarse, and BB aerosols in this site using the new hybrid
algorithm. These aerosol types could be from the Chihuahuan Desert, an inland desert
covering 12% of Mexico's area and a major source of coarse and dust aerosols.



Additionally, the literature shows that Mexico City is surrounded by forested
mountains, which experience many wildfires during the dry period between
November and May; this accounts for BB aerosols in Mexico City (Yokelson et al.
2007). Finally, the BB aerosols identified at the Moldova site could be attributed to its
rich vegetation cover.

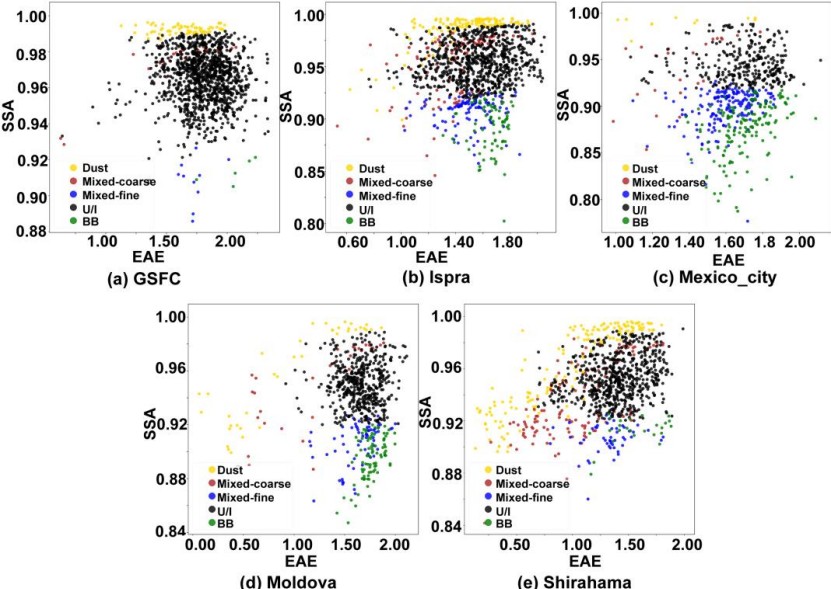


**Figure 10**. Same as Figure 9 but for urban/industrial aerosol.

### 4.2.4 Biomass burning aerosol

The selected sites were mainly located in the mountains and highlands. Figure 11
shows the aerosol types identified using the new hybrid algorithm. Large amounts of
BB aerosols were identified at all sites. Additionally, a small amount of dust and
mixed-coarse aerosols were identified at the Alta Floresta site, transported over a long
distance from the Patagonian Desert in Argentina, in southern South America.
Moreover, the city where the site is located is industrially developed and has a large
population; therefore, more U/I aerosols were identified using the new hybrid
algorithm. The geographically close Abracos Hill and Alta Floresta sites were
characterized by the same aerosol type and source. Furthermore, one data point in
Lake Argyle was classified as a dust aerosol. This means that, although the site is



located on the Kimberley Plateau, Australia has a large desert area, and coarse
aerosols still exist. Lastly, a few U/I and several dust-type aerosols were identified at
the Mongu site, possibly caused by aerosol emissions from nearby cities and dust
transport from the Saharan Desert.

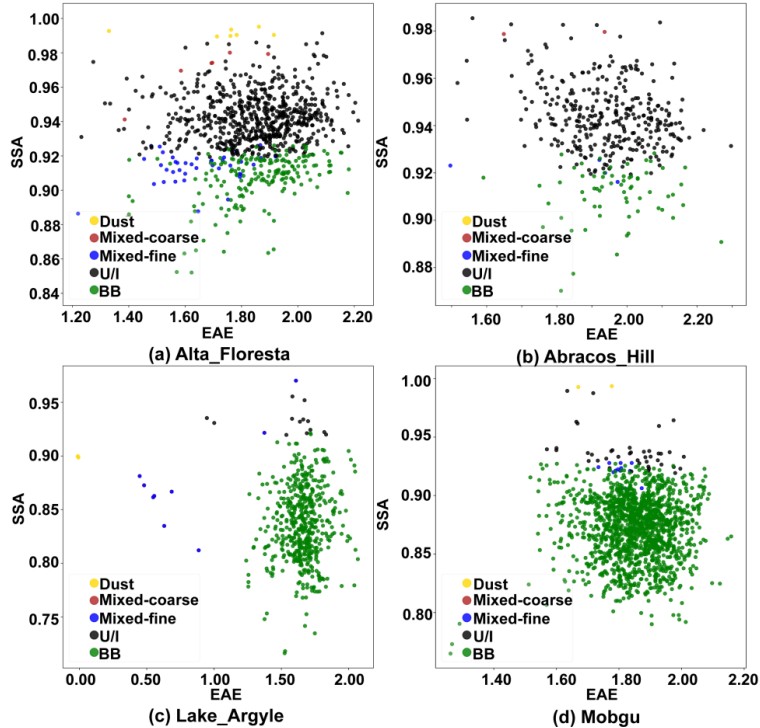


**Figure 11**. Same as Figure 10 but for BB aerosol.

## 4.3 Aerosol type distribution on a global scale

Given the advantages and accuracy of the new hybrid algorithm in identifying
aerosol types, we used it to divide the data of AERONET sites in different regions of
the world to obtain global aerosol type distribution information. The aerosol types of
each continent are shown in Figures 12-16. Additionally, Figure 17 shows the global
aerosol-type distribution. Notably, the pie chart was placed on each site in the study,
which is a "point source" assessment of the aerosol type and does not represent the
entire region (the size of the pie chart is independent of the optical properties).
Moreover, the sites were screened, and only those with valid data of > 100 aerosol



types were considered; however, offshore sites and sites classified as marine aerosol-
dominated by other literature were excluded.
Figure 12 shows pie charts of the aerosol types for each scanned AERONET site
in North America. The U/I aerosols, particularly in most mid-eastern regions,
contained mixed and small amounts of biomass aerosols. Additionally, the AERONET
sites in large cities, such as Chicago, New York, Toronto, Ottawa, and Boston, had U/I
aerosols. Many studies have shown that dust aerosols from the Saharan Desert can
cross the Atlantic Ocean to North America in summer. Moreover, there is an inland
desert in western North America, the Chihuahua Desert, responsible for a small
amount of dust and mixed aerosols at the AERONET sites in North America.
Additionally, wildfires in western North America and household wood burning
contribute to most BB aerosols yearly. The central region site is affected by the
environment, with an increased proportion of BB aerosols, and U/I aerosols are still
prevalent because the site is located in a large city and is densely populated.

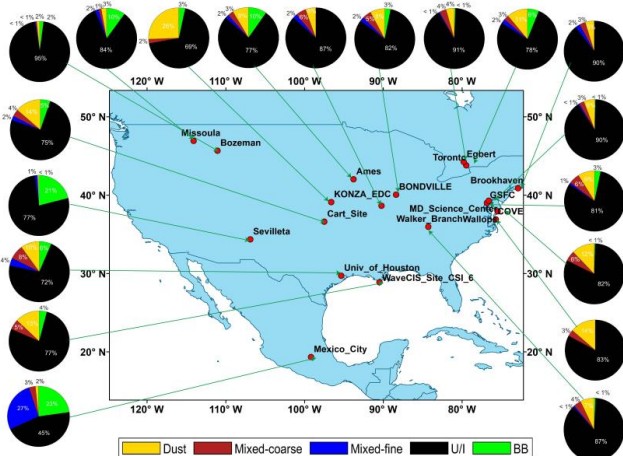

**Figure 12**. Pie charts of the aerosol types at the major sites of North American.
Figure 13 shows the aerosol types in Africa. Northern Africa has the largest desert
in the world, the Saharan Desert; therefore, dust aerosols dominate north of the
equator in Africa. However, some AERONET sites in the Sudanese steppe were
primarily BB, with some U/I aerosols in nearby urban sites. The Ilorin site is a typical





mixed aerosol site close to the equator with a small amount of BB aerosols. Most sites

close to the Atlantic coast were affected by dust aerosols, even those on the islands of

Cape Verde. The reliability of the new model in distinguishing U/I and BB aerosols is

demonstrated. Sites in Southern Africa, such as Namibia, Botswana, and Zambia, are

dominated by BB aerosols. Nevertheless, studies have shown the presence of U/I

aerosols at sites in the urban areas of South Africa. Although U/I and BB aerosols are

difficult to distinguish, the two can be identified in the context of a large urban

population and less biomass combustion, thus establishing the model's accuracy.

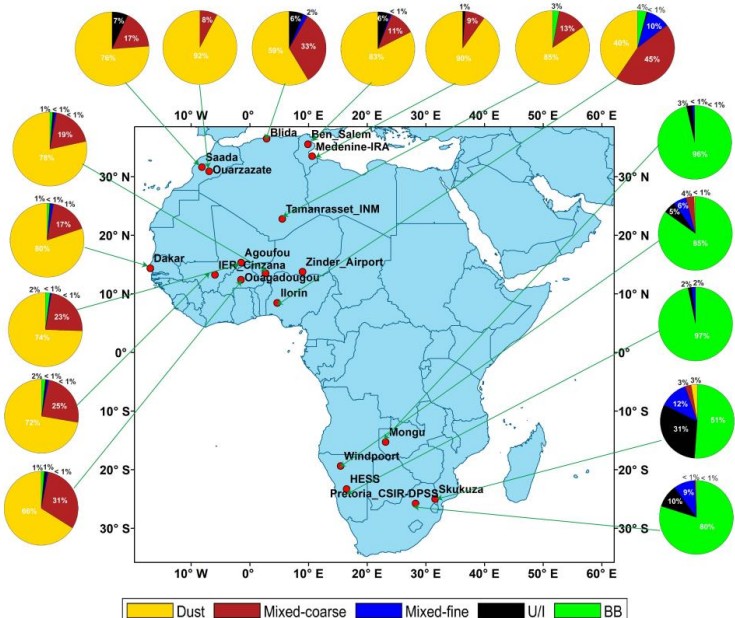

**Figure 13**. Same as Figure 12 but for Africa.

The aerosol types in South America are shown in Figure 14. Here, only eight sites

met the requirement for valid data >100 aerosol types. South America is mainly

dominated by mountainous plateaus, and under the influence of the Brazilian warm

current, many tropical rainforests are distributed in the south; therefore, the

background aerosols are mainly BB aerosols. As shown in Figure 14, large cities, such

as Rio Branco, Campo Grande, Manaus, Santa Cruz, and São Paulo, showed an

increased proportion of anthropogenic and mixed aerosols because of their large





population and developed industries. Due to the tropical rainforest climate in southern
South America, the proportion of BB aerosols increased, such as that at the Cuiaba
site near the Amazon River. Additionally, the Manaus site contained a small amount of
dust aerosols that were presumably transported across the Atlantic Ocean from
African dust at the same latitude.

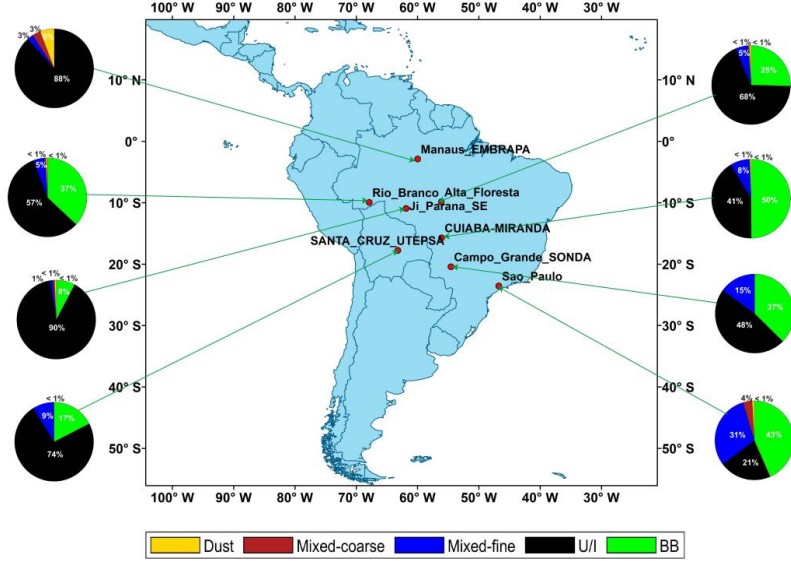


**Figure 14**. Same as Figure 12 but for South America.
The aerosol types in Asia are shown in Figure 15. In western Asia, influenced by
the Indian Desert, sites on the Indian Peninsula were dominated by coarse-particle
aerosols, including dust and mixed coarse aerosols. Kanpur and Pune are densely
populated cities in India, with more mixed-fine aerosols produced by human
activities. Additionally, in Southeast Asia, all sites contained BB aerosols, consistent
with Hamill (2014). This is because of the abundance of tropical rainforests in
Southeast Asia. Moreover, some urban sites, such as Singapore and Penang, had large
numbers of U/I and mixed-fine aerosols. The coastal areas of East Asia, which are
densely populated and industrially developed, were mainly dominated by U/I
aerosols. Moreover, dust aerosols appeared at these sites due to dust transported from
the Taklamakan Desert in East Asia.

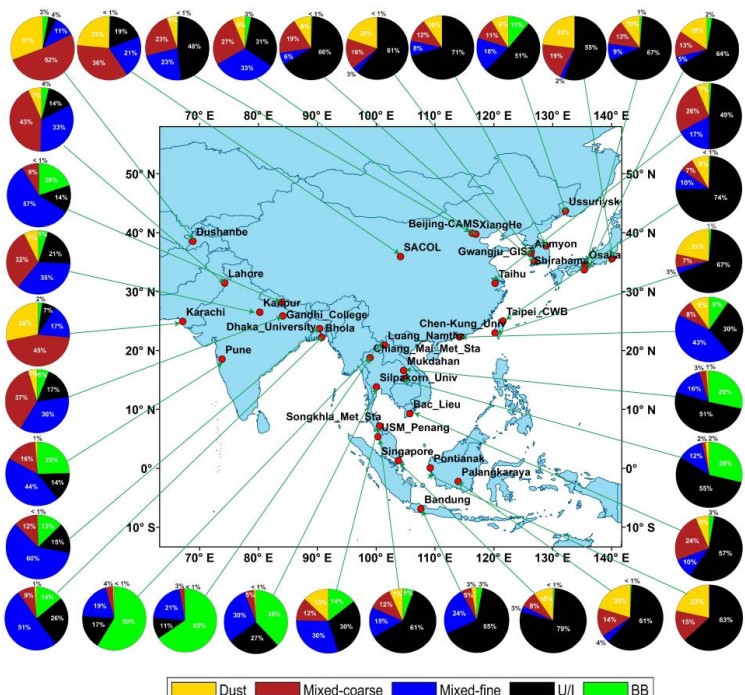

**Figure 15**. Same as Figure 12 but for Asia.

The inland areas of East Asia have a smaller population than the coastal areas;
therefore, the proportion of U/I aerosols was small, and that of mixed aerosols was
high. Generally, mixed aerosols are more easily overestimated than U/I aerosols;
however, the new hybrid algorithm identified a larger proportion of U/I aerosols than
mixed aerosols at Asian sites. Therefore, this new hybrid algorithm can be considered
for improving the classification of mixed aerosols versus U/I aerosols.

Similarly, southern Europe, which is close to the Saharan and Arabian deserts,
was dominated by dust aerosols, with small amounts of mixed and U/I aerosols.
Northern European sites have many cities and a large population; therefore, the
aerosol type was mainly U/I aerosols, identified using the new hybrid algorithm
(Figure 16). Additionally, small amounts of BB aerosols were identified at most sites
in Europe because of olive groves in agricultural lands in the EU, which produce 91%
of the world's olive oil (Lopez-Pineiro et al., 2011). Papadakis et al. (2015) suggested
that the biomass produced from olive oil is used for heating and industry, and its
combustion produces carbonaceous aerosols, considered the major source of fine



particle aerosols in Europe during winter (Puxbaum et al., 2007).

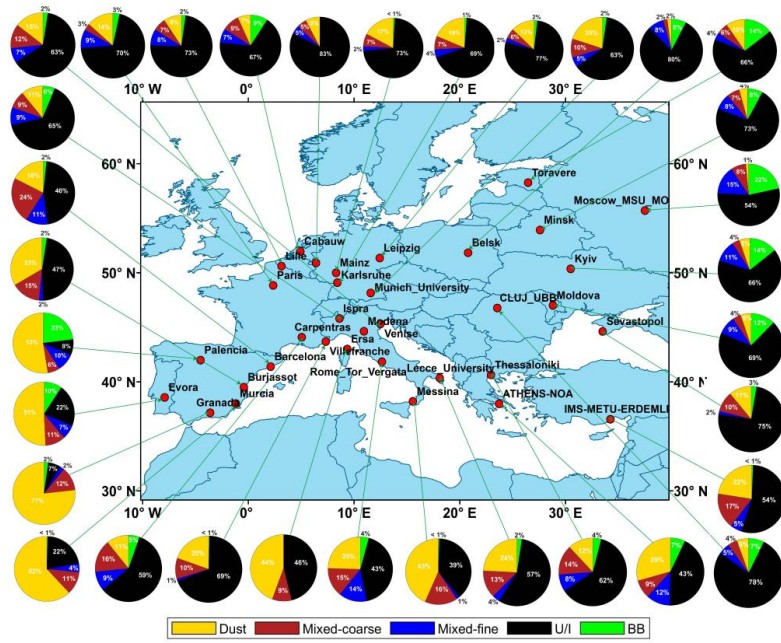


**Figure 16**. Same as Figure 12 but for Europe.

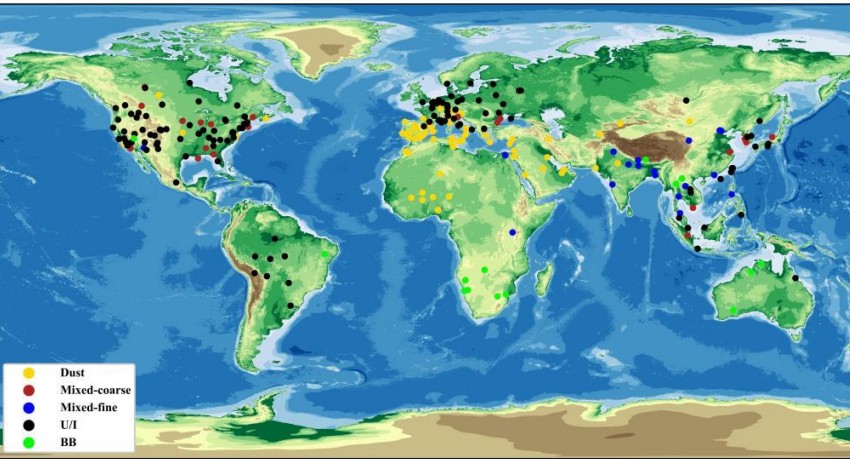


**Figure 17**. Global dominant aerosol types distribution based on AERONET sites.

The global distribution of dominant aerosols in the AERONET site is shown in

Figure 17. The graph does not include marine aerosols. There are more aerosol sites
on the global map than those on each continent because AERONET sites with > 5



years of data were selected for the global map; however, sites with > 100 valid data
points were required for each continent. The global distribution map shows that many
BB aerosols were distributed between 20°N and 20°S. This is because this region has
a predominantly tropical rainforest climate, with many tropical rainforests and more
carbon-containing aerosol emissions. This finding is consistent with those from
previous studies that found that global BB aerosols mainly originate from Africa
(approximately 52%), followed by South America (approximately 15%), equatorial
Asia (approximately 10%), boreal forests (approximately 9%), and Australia
(approximately 7%) (Van G. R. et al., 2010). Furthermore, the global distribution map
shows a clear distribution band of dust aerosols between 5°N and 35°N, originating
from the Saharan Desert in Africa and the Saudi Arabian Desert in Western Asia,
which are transported across the ocean to other regions.

## 5. Conclusion

We developed a new hybrid algorithm to support the rapid classification of
aerosol types by building an aerosol optical database for global AERONET sites. This
hybrid algorithm is a complex aerosol-type processing algorithm that effectively
integrates machine learning and density clustering algorithms. Additionally, this
algorithm is not limited by the amount of data and improves the accuracy of aerosol-
type classification. On investigating the aerosol types at specific sites with dominant
aerosols, we observed that different sites contained one or more aerosol types, with
the composition of some specific dominant aerosol sites being more complex than that
of others. The new algorithm showed a higher accuracy than that shown by algorithms
used in previous studies in identifying aerosol types at specific sites, particularly in
distinguishing between U/I and mixed-fine aerosols. Finally, the recognition results of
the new hybrid algorithm were closer to the baseline CRI, confirming that the hybrid
algorithm is better than the density-clustering algorithm. On investigating the aerosol
types at global sites across the continents using the new algorithm, we observed the
dominance of different types of aerosols at different sites, and the composition of



these could be logically and effectively attributed to the geographical location, energy
consumption structure, meteorological conditions and activities happening at the
respective sites.
In this study, the existing aerosol type identification algorithm was improved
using global ground-based AERONET optical property parameter data, and the spatial
distribution characteristics of global aerosol types were analyzed, which impacted
aerosol radiation research and optical thickness inversion accuracy. However, marine
aerosols were not considered in this study. This is a limitation of our study, and further
studies are required to include the optical properties of marine aerosols in model
building.
**Author contributions**
**Feng Zhang** designed the study. **Xiaoli Wei** analyzed the results, and wrote the
original draft. **Qian Cui** collected and processed the data. **Leiming Ma** revised the
paper and given constructive suggestions. **Wenwen Li** constructive comments on the
paper. **Peng Liu** revised the paper. All authors contributed to the study.
**Competing interests**
The authors declare that they have no conflict of interest.
**Acknowledgments**
This work was supported by the National Key R&D Program
(2021YFB3900401), the National Natural Science Foundation of China (42105081
and 42075125) and Science and Technology Foundation of Shanghai (23ZR1454100)

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
