# Peer review of "Global aerosol typing classification using a new hybrid"

_EGUsphere, 2023_

## Referee Comment (RC2)

The manuscript by Xiaoli Wei et al., attempts to develop a new aerosol type classification model using an innovative hybrid algorithm. An optical database was developed using Mie scattering and a complex refractive index, different aerosol types were identified by applying a random forest algorithm trained on aerosol optical parameters obtained from the AERONET sites. The method effectively integrates machine learning and density clustering algorithms and is not limited by the amount of data, improving the accuracy of aerosol type classification. The study shows good consistency between the new method and traditional Gaussian density cluster method. In conclusion, this manuscript is logically organized and well written, and provides some insights and new methods to identify aerosol types based on aerosol optical parameters. However, a number of issues need to be addressed before it can be accepted by STOTEN.

Major comments

1. Line 134-136, Table 1 and Figure 1: Site names need to be aligned, e.g., "AltaFloresta" and "Alta_Floresta", "Abracos Hill" and "Abracos_Hill", etc.

2. Line 160-161: It is mentioned here that for the first time the median value of the complex refractive index is used as a criterion for identifying the type of aerosol. What are the advantages of applying the median value of the complex refractive index? It is suggested that a specific explanation be given in the manuscript.

3. Line 183: Punctuation is in red.

4. Line 193: Punctuation is in blue.

5. Line 220-227: The manuscript mentioned the use of aerosol size distribution parameters, CRI and Mie scattering model to build a database of aerosol optical parameters instead of using AERONET data directly. Perhaps the manuscript could give more explanation and how the accuracy of the established database can be verified.

6. It would be helpful to explain more clearly why the Random Forest model was chosen over other AI models, and the specific steps in the implementation of the Random Forest model.

7. It is recommended to add a comparison with traditional aerosol type identification algorithms to highlight the advantages of your own hybrid algorithm.

8. Figure 7: Perhaps adjusting the horizontal and vertical coordinates of each sub-figure to place the image horizontally would make it easier for the reader to see.

9. Figure 12-16: Pie charts labelled with illegible fonts.

---

## Author Comment (AC4)

**Response to Reviewer #2:**

The manuscript by Xiaoli Wei et al., attempts to develop a new aerosol type classification model using an innovative hybrid algorithm. An optical database was developed using Mie scattering and a complex refractive index, different aerosol types were identified by applying a random forest algorithm trained on aerosol optical parameters obtained from the AERONET sites. The method effectively integrates machine learning and density clustering algorithms and is not limited by the amount of data, improving the accuracy of aerosol type classification. The study shows good consistency between the new method and traditional Gaussian density cluster method. In conclusion, this manuscript is logically organized and well written, and provides some insights and new methods to identify aerosol types based on aerosol optical parameters. However, a number of issues need to be addressed before it can be accepted by STOTEN.

1. Line 134-136, Table 1, and Figure 1: Site names need to be aligned, e.g., "AltaFloresta" and "Alta_Floresta", "Abracos Hill" and "Abracos_Hill", etc.

**Response:** Thank you for your reminder.

We have thoroughly examined our paper and confirmed that the names of all sites are consistent, with a focus on alignment in Table 1 and Figure 1. Please refer to the new contents from line 137-151.

2. Line 160-161: It is mentioned here that for the first time the median value of the complex refractive index is used as a criterion for identifying the type of aerosol. What are the advantages of applying the median value of the complex refractive index?

It is suggested that a specific explanation be given in the manuscript.

**Response:** Thanks for the constructive comment.

The reasons to apply the median value of the complex refractive index (CRI) as a baseline to determine which algorithm performs better are added on page 7, line 161-171.

- The CRI is a key microphysical parameter of aerosols. It determines the inherent optical properties of aerosols, such as scattering and absorption (Raut and Chazette,

2008). It is vital for determining aerosols' chemical and physical compositions (Dubovik and King, 2000) and the CRI value is known for pure aerosol components (Nandan et al., 2021).

- Unlike the mean, the median CRI value is employed in this research for it represents the central tendency of data, especially beneficial in skewed distributions or when outliers are present. This is particularly useful when an average value of a specific aerosol-type might be influenced by the presence of other aerosol types.

In addition, the advantages of applying the median value of the CRI are also described on page 13 (line 284-290).

The CRI is an inherent optical property of aerosols. Aerosols in the real atmosphere are usually mixed with different types of particles, which a single refractive index cannot identify; however, the CRI represents the entire aerosol model in the atmosphere (Redemann et al., 2000). Ideally, the CRI and aerosol components can be mutually determined (Wu et al., 2021). The CRI can effectively characterize the main properties of the aerosols and accurately quantify the difference between aerosol-type identification algorithms.

3. Line 183: Punctuation is in red.

**Response:** Thanks for the reminder. The mistake has been corrected on line 311.

4. Line 193: Punctuation is in blue.

**Response:** Thank you very much. It has been corrected on line 315. We have carefully checked the whole paper and cleaned up the writing issues in the revised version.

5. Line 220-227: The manuscript mentioned the use of aerosol size distribution parameters, CRI and Mie scattering model to build a database of aerosol optical parameters instead of using AERONET data directly. Perhaps the manuscript could give more explanation and how the accuracy of the established database can be verified.

**Response:** Thanks for the constructive comment. In the new manuscript, Mie scattering model is further explained (see page 11 line 240-245 and page 12 line 255-259):

Mie scattering model is applied to reconstruct the aerosol optical properties parameters. Theoretically, it is generally used for describing the analytic solution to

Maxwell's equations for light scattering from spherical particles. It works well in describing aerosol scattering and absorbing properties in the atmosphere and is the basis of radiative transfer, Lidar, and optical particle characterization (Michael et al., 1994).

In this study, the size distribution parameters and the median value of complex refractive index obtained from ground AERONET sites were used as input parameters of Mie scattering model to calculate the normalized-AOD, SSA, g, and EAE. Many studies have proven it's reliable to calculate these optical parameters (Ma et al., 2007; Bian et al., 2017). Specifically, Zhao et al. (2020) characterized the diurnal variations of asymmetry factor (g) with Mie scattering model and field measurements experiments to prove it truly reflects its characteristics. Nandan et al. (2021) pointed out that the complex refractive index of optically effective aerosol can be estimated by using Mie algorithm and well matched with observational results. Fu et al. (2009) simulated the single-scattering properties of non-spherical dust aerosols and indicated that the relative errors in global reflectively caused by simulated single-scattering properties are always less than 5%. They also explained that the assumption of homogeneously mixed particles in Mie theory and sphericity shape contributes to these errors. Quirantes et al. (2019) calculated extinction-related angström exponent based on a lower value of the size distribution and suggested that the Mie scattering model is a more reasonable model with the advantage of lower computing load.

Therefore, Mie scattering model is a mature model, that can be used to establish an optical parameter database accurately and reliably.

Reference:

Ma Lin.: Measurement of aerosol size distribution function using Mie scattering - Mathematical considerations., Journal of aerosol science, 38(11),1150-1162, https://doi.org/10.1016/j.jaerosci.2007.08.003, 2007.

Zhao, G., Li, F., & Zhao, C.: Determination of the refractive index of ambient aerosols. Atmospheric Environment, 240, 117800. https://doi.org/10.1016/j.atmosenv.2020.117800,2020

Bian, Yuxuan et al.: Development and Validation of a CCD-Laser Aerosol Detective System for Measuring the Ambient Aerosol Phase Function., Atmospheric measurement techniques, 10 (6),2313–2322. https://doi.org/10.5194/amt-10-2313, 2017

Nandan, R., Ratnam, M.V., Kiran, V.R., Madhavan, B.L., Naik, D.N.: Estimation of Aerosol Complex Refractive Index over a tropical atmosphere using a synergy of in-situ measurements., Atmospheric Research, 257, 105625, ttps://doi.org/10.1016/j.atmosres.2021.105625, 2021.

Fu, Q., Thorsen, T.J., Su, J., Ge, J., & Huang, J.: Test of Mie-based single-scattering properties of non-spherical dust aerosols in radiative flux calculations. Journal of Quantitative Spectroscopy & Radiative Transfer, 110, 1640-1653. https://doi.org/10.1016/j.jqsrt.2009.03.010,2009

Quirantes, Arturo, et al.: Extinction-related Angström exponent characterization of submicrometric volume fraction in atmospheric aerosol particles., Atmospheric Research, 228(D24), 270-280, https://doi.org/10.1016/j.atmosres.2019.06.009,2019

6. It would be helpful to explain more clearly why the Random Forest model was chosen over other AI models, and the specific steps in the implementation of the Random Forest model.

**Response:** Thank you for your advice.

To enhance the algorithm's adaptability, the implementation in step 3 is not rigidly confined to the random forest model. It can be substituted with any other classification machine learning algorithm. In the revised manuscript, we provided an explanation for selecting the random forest algorithm (see page 15, line 324-336).

The model has a high prediction accuracy, excellent tolerance for abnormal values and noise, and a hard overfit. In a comparison by Fernandez (2014), the random forest algorithm ranked as the top performer among 179 classification algorithms. In addition, the evaluation matrix was brought into this study, and it further quantitatively assesses the performance of the Gaussian density clustering algorithm and the new hybrid algorithm. The metric indexes include accuracy, recall, precision, and F-scores (Reddy et al., 2022). Here, the indexes are adjusted to micro-precision, micro-recall, micro-F1-score, and accuracy to solve the multi-classification problem. Micro refers to the weighted average of the five aerosol types rather than the arithmetic mean, due to the large difference in sample size among the five aerosol types, the arithmetic mean is highly susceptible to the influence of very large or very few sample size aerosol types.

7. It is recommended to add a comparison with traditional aerosol-type identification algorithms to highlight the advantages of your own hybrid algorithm.

**Response:** Thank you for your copious and constructive review.

The contrast between the conventional Gaussian density clustering aerosol identification algorithm and the novel hybrid algorithm is detailed in section 4.1, and we have further supplemented the content in this updated manuscript:

In addition, the evaluation matrix was brought into this study, and it further quantitatively assesses the performance of the Gaussian density clustering algorithm and the new hybrid algorithm. The metric indexes include accuracy, recall, precision, and F-scores (Reddy et al., 2022). Here, the indexes are adjusted to micro-precision, micro-recall, micro-F1-score, and accuracy to solve the multi-classification problem. Micro refers to the weighted average of the five aerosol types rather than the arithmetic mean, due to the large difference in sample size among the five aerosol types, the arithmetic mean is highly susceptible to the influence of very large or very few sample size aerosol types (see page 15, line 327-336 ).

Table 5 shows the metric index value of the random forest algorithm in the new hybrid algorithm. The micro-precision, micro-recall, micro-F1-score, and accuracy are 0.95, 0.89, 0.91, and 0.89, respectively. The values in the table are calculated based on the core values of the window obtained by the Gaussian density clustering algorithm. Therefore, these high indicators further verify the reliability of the new hybrid algorithm (see p.17, line 371-378).

Table 5 Matrix evaluation between the new hybrid classification algorithm and the Gaussian density clustering algorithm

|  | Micro-Precision | Micro-Recall | Micro-F1-Score | Accuracy |
|---|---|---|---|---|
| New Hybrid algorithm | 0.95 | 0.89 | 0.91 | 0.89 |

8. Figure 7: Perhaps adjusting the horizontal and vertical coordinates of each sub-figure to place the image horizontally would make it easier for the reader to see.

**Response:** Thank you for your suggestion. The figure has been modified from a horizontal to a vertical orientation. Additionally, the resolution of the images in this manuscript has been enhanced to ensure clarity and readability. (see Figure 7).

9. Figure 12-16: Pie charts labeled with illegible fonts.

**Response:** Thank you for your advice. we have revised the pie charts to ensure the numbers are clearly visible and easy to read (see Figure 12-16).

---

## Author Comment (AC5)

**Response to Reviewer #1:**

The manuscript" Global aerosol typing classification using a new hybrid algorithm utilizing Aerosol Robotic Network data" aims to develop a new aerosol-type classification model using an innovative hybrid algorithm to improve the precision and efficiency of aerosol-type identification. The study shows good consistency between the new method and traditional Gaussian density cluster method, with consistency rates of 90%, 85%, 84%, 84%, and 100% for dust, mixed-coarse, mixed-fine, urban/industrial, and biomass burning aerosols, respectively. Overall, the manuscript provides a well-structured and clear overview of the study design, methodology, and results. The authors communicate the significance of their findings in addressing the issue of classifying aerosol type accurately and efficiently in global scale, which has important implications for aerosol inversion and aerosol pollution study. However, there are a few areas where the manuscript could be improved.

1. The authors should elaborate on why they chose Mie scattering model to build an aerosol optical database for classifying the aerosol type.

**Response:** Thank you for your constructive advice. The Mie scattering model, known for its simplicity and practicality, provides an analytic solution to Maxwell's equations for light scattering by ideal spherical particles. It efficiently depicts the scattering and absorption properties of aerosols in the atmosphere, serving as fundamental basis of radiative transfer, Lidar, and optical particle characterization (Ma et al.,2007; Bian et al., 2017; Michael et al., 1994) (see line 240-245).

There are two reasons why the Mie scattering model is used to calculate aerosol optical parameters instead of directly using observational data: First, the optical parameters from AERONET site, such as AOD and complex refractive index, are seriously missing, and the effective information provided is limited, particularly, the amount of observation data of AERONET site dominated by biomass combustion can no longer meet the requirement of machine learning algorithms. Second, the calculation of aerosol optical properties by fixing the refractive index of the Mie scattering model

allows for precise determination of aerosol types (see line 231-237 ).

2. It would be helpful to provide more context on the limitations of their approach and future directions for research in this area.

**Response:** Thanks for your advice. On page 32, the last paragraph, we further enriched the limitations and pointed out the possible future directions for the aerosol-type identification study.

However, marine aerosols were not considered for fewer valid data after site screening in this study. The dust aerosol in the Mie scattering model is assumed to be spherical, and the actual natural environment is not spherical, which inevitably brings errors, and the accuracy of the optical database needs to be further improved. In the future, with the development of machine learning, random forest algorithms can be replaced with stronger learning algorithms. Meanwhile, multi-source satellite data and reanalysis products can be incorporated into aerosol-type identification. This study will provide support for the identification and control of air pollution sources. (see line 656-661).

3. The manuscript needs to have more information in the result about the improvements in calculation time efficiency of aerosol type classification with a specific scale.

**Response:** Thanks for the advice.

Additionally, in this study, the number of 326400 data points from optical parameters database and 98000 observed data for calculation spans from Jan.1st,1993 to Dec.31st,2021, passing through Gaussian kernel density clustering algorithm and new hybrid algorithm Python progresses, which is archived on the personal Windows system computer (Intel® Core™ i7-10710U,16G DDR4 2666MHz, 512G PCIE SSD). The computational time for the two algorithms indicates the new hybrid algorithm runs faster than the Gaussian kernel density clustering algorithm with huge quantities of data and trained in advance, which can obtain aerosol type in 20 seconds, in contrast, it will take 30 to 40 seconds to obtain aerosol type in one site by using the Gaussian algorithm (see line 449-458 ).

4. The manuscript requires a clearer explanation of how the random forest model was implemented and any potential biases associated with the model.

**Response:** Thanks. On page 15, line 321-336, and page 17, line 371-379, we implement the further assessment of the random forest algorithm in the new hybrid algorithm and analyze the potential biases associated with the model.

In this study, the evaluation matrix was brought into this study, and it further quantitatively assesses the performance of the Gaussian density clustering algorithm and the new hybrid algorithm. The metric indexes include accuracy, recall, precision, and F-scores (Reddy et al., 2022). Here, the indexes are adjusted to micro-precision, micro-recall, micro-F1-score, and accuracy to solve the multi-classification problem. Micro refers to the weighted average of the five aerosol types rather than the arithmetic mean, due to the large difference in sample size among the five aerosol types, the arithmetic mean is highly susceptible to the influence of very large or very few sample size aerosol types.

Table 5 shows the metric index value of the random forest algorithm in the new hybrid algorithm. The micro-precision, micro-recall, micro-F1 score, and accuracy are 0.95, 0.89, 0.91, and 0.89, respectively. These metrics are derived from the core values of the window, as determined by the Gaussian density clustering algorithm. Consequently, the strong performance of these indicators further confirms the efficacy and reliability of the newly developed hybrid algorithm.

Table 5 Matrix evaluation between hybrid classification algorithm and Gaussian density clustering algorithm

|  | Micro-Precision | Micro-Recall | Micro-F1-Score | Accuracy |
| --- | --- | --- | --- | --- |
| New Hybrid algorithm | 0.95 | 0.89 | 0.91 | 0.89 |

Line 99-106 describe the limitation of the machine learning algorithm, which also the potential bias of random forest. The content is as follows:

However, some challenges remain in identifying aerosol types through machine learning. First, the amount of valid ground aerosol property data that can be used for training is less due to cloud removal and quality control. Second, the accuracy of machine learning depends on the labeled aerosol typing dataset, and finding a suitable

classification method to classify the dataset is challenging. Third, evaluating the accuracy of the final trained model is also tedious (Zhang & Li, 2019; Siomos et al., 2020; Choi, et al., 2021a,b).

In addition, the random forest algorithm associated with learning ability is insufficient and can be replaced by other stronger learning algorithms in the future (see line 656-658).

5. More relevant literature review should be included, especially those from the last three years.

**Response:** Thank you for your patience. We added several recent three years of study in this field to support our references. Please, refer to these references:

Elham Ghasemifar: Climatology of aerosol types and their vertical distribution over Iran using CALIOP dataset during 2007–2021,Remote Sensing Applications: Society and Environment,32, 101053, 2352-9385,https://doi.org/10.1016/j.rsase.2023.101053.2023.

Nandan, R., Ratnam, M.V., Kiran, V.R., Madhavan, B.L., & Naik, D.N.: Estimation of Aerosol Complex Refractive Index over a tropical atmosphere using a synergy of in-situ measurements. Atmospheric Research, 257, 105625, https://doi.org/10.1016/J.ATMOSRES.2021.105625, 2021

Reddy LA, Glover TA, Dudek CM, Alperin A, Wiggs NB, Bronstein B.: A randomized trial examining the effects of paraprofessional behavior support coaching for elementary students with disruptive behavior disorders: Paraprofessional and student outcomes. J Sch Psychol. 2022 Jun;92:227-245. https://doi.org/10.1016/j.jsp.2022.04.002, 2022.

Wang J, Liu Y, Chen L, Liu Y, Mi K, Gao S, Mao J, Zhang H, Sun Y, Ma Z.: Validation and calibration of aerosol optical depth and classification of aerosol types based on multi-source data over China. Sci Total Environ. 2023 Dec 10;903:166603. doi: 10.1016/j.scitotenv.2023.